# Dynamic Characteristics Prediction Model for Diesel Engine Valve Train Design Parameters Based on Deep Learning

Wookey Lee [1,2], Tae-Yun Jung [2] and Suan Lee [3,*]

1 Biomedical Science and Engineering, Inha University, Incheon 22212, Republic of Korea; trinity@inha.ac.kr
2 Department of Industrial Engineering, Inha University, Incheon 22212, Republic of Korea
3 School of Computer Science, Semyung University, Jecheon 27136, Republic of Korea
* Correspondence: suanlee@semyung.ac.kr; Tel.: +82-43-649-1273

**Abstract:** This paper presents a comprehensive study on the utilization of machine learning and deep learning techniques to predict the dynamic characteristics of design parameters, exemplified by a diesel engine valve train. The research aims to address the challenging and time-consuming analysis required to optimize the performance and durability of valve train components, which are influenced by numerous factors. To this end, dynamic analyses data have been collected for diesel engine specifications and used to construct a regression prediction model using a gradient boosting regressor tree (GBRT), a deep neural network (DNN), a one-dimensional convolution neural network (1D-CNN), and long short-term memory (LSTM). The prediction model was utilized to estimate the force and valve seating velocity values of the valve train system. The dynamic characteristics of the case were evaluated by comparing the actual and predicted values. The results showed that the GBRT model had an $R^2$ value of 0.90 for the valve train force and 0.97 for the valve seating velocity, while the 1D-CNN model had an $R^2$ value of 0.89 for the valve train force and 0.98 for the valve seating velocity. The results of this study have important implications for advancing the design and development of efficient and reliable diesel engines.

**Keywords:** diesel engine; valve train dynamics; deep learning; GBRT; DNN; LSTM; 1D-CNN

## 1. Introduction

Artificial intelligence (AI) has emerged as a pivotal technology in the Fourth Industrial Revolution, gaining prominence across several domains such as manufacturing, finance, and healthcare. The sub-technologies of AI, namely machine learning and deep learning, have become increasingly ubiquitous in various fields, including computer vision [1], language models [2], speech recognition [3], and industrial fault diagnosis [4]. As a result, AI has garnered considerable attention as a transformative force in advancing these fields, holding immense potential for revolutionizing the industry and enhancing human capabilities.

Due to their high performance, the dynamic characteristics of the design parameters of valve train cases have emerged as a crucial research area. In the case study, a diesel engine valve train was selected, because valve trains have not only directly affected the engine performance but also caused mechanical wear and noise generation due to repeated motion [5]. Furthermore, the acceleration of valve movement increases as the engine speed increases. However, high acceleration leads to an increase in the inertial force of the valve train, which can result in abnormal dynamic behaviors such as cam follower jumping, a high valve seating speed, valve bouncing, among others, adversely impacting the dynamic performance of the valve train [6]. To maximize engine performance while ensuring the dynamic performance and durability of the valve train, it is imperative to select cam shapes and components while considering the dynamic characteristics of the valve train. This study aims to address this issue by utilizing machine learning and deep learning techniques to predict valve train dynamics and select optimal specifications.

The aim of this research is to investigate the feasibility of utilizing machine learning and deep learning techniques to select the most suitable specifications that meet the required dynamic performance and durability for designing valve train components in diesel engines. The behavior of a valve train system is influenced by various factors, including engine rotational speed, engine valve spring constant, cam profile, and the mass and rigidity of the components that make up the valve train system. Analyzing the design and interpretation of the valve train system that requires new or modified development currently takes a considerable amount of time due to these numerous factors. Therefore, by establishing a prediction model using machine learning and deep learning techniques, the researchers aim to develop a tool that can swiftly assess the applicability of component specifications by verifying the dynamic characteristics of the valve train system with simple input data.

The main contributions of this paper are as follows:

1.  We compare various machine learning and deep learning models such as the gradient boosting regressor tree (GBRT), the deep neural network (DNN), the one-dimensional convolution neural network (1D-CNN), and the long short-term memory (LSTM) to predict the dynamic characteristics of diesel engine valve train design parameters.
2.  We present an account of the data preprocessing process, which involved importing raw data, creating a dataset with speed and application columns, aligning lift axes, removing unnecessary data, and classifying specifications for model training and validation.
3.  We demonstrate the validity and effectiveness of this study by performing a detailed comparative analysis of models predicting valve train force and valve seating velocity over a range of crankshaft angles and engine speeds.

The structure of this paper is outlined as follows. In Section 2, the background and related work are reviewed. Section 3 describes the prediction model in detail. Section 4 details the data engineering and model construction, while Section 5 presents the experimental analysis results. Section 6 describes the discussion and limitations of this paper, Finally, Section 7 concludes the paper.

## 2. Background and Related Work

### 2.1. Valve Train

The internal combustion engine is an essential power source in various fields, including transportation, industrial machinery, and power generation. In a four-stroke internal combustion engine, the valve train plays a vital role in regulating the intake and exhaust processes, which are essential for generating power. The valve train system is responsible for controlling the opening and closing of the intake and exhaust valves during the suction, compression, explosion, and exhaust processes. For diesel engines, the intake and exhaust valves are crucial components that contribute significantly to the combustion chamber's air intake and exhaust gas expulsion during the suction and exhaust processes, respectively. As illustrated in Figure 1, the valve train system includes several mechanical elements such as a camshaft, tappet, pushrod, rocker arm, valve spring, and valve seat, which work in coordination to ensure the proper operation of the valves in accordance with the engine's crankshaft rotation.

The valve train system's performance significantly affects the engine's performance, making it a critical system in the internal combustion engine. The valve train can be broadly classified into two types: overhead camshaft (OHC) and overhead valve (OHV). This study focuses on the OHV-type valve train, which is prevalent in diesel engines. To ensure the proper operation of the valve train system, it is essential to consider the dynamic characteristics of the components. High acceleration can lead to an increase in the inertial force of the valve train, which can result in abnormal dynamic behaviors such as cam follower jumping, high valve seating speed, and valve bouncing, among others, adversely impacting the valve train's dynamic performance. Therefore, it is critical to select cam shapes and components while considering the dynamic characteristics of the valve train to ensure maximum engine performance and the valve train's dynamic performance and durability.

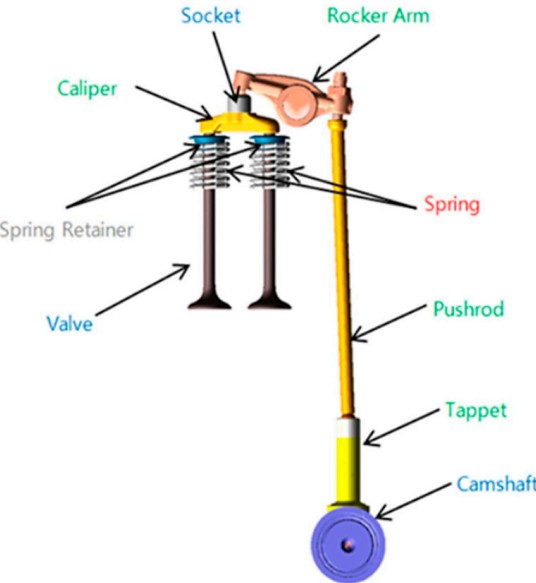

**Figure 1.** Schematic diagram of the diesel engine valve train.

This study aims to predict the dynamic characteristics of the valve train system using machine learning and deep learning techniques to develop a tool that can swiftly assess the applicability of component specifications with simple input data.

### 2.2. Dynamic Characteristics of Diesel Engine Valve Trains

The abnormal dynamic properties of the valve train significantly impact the engine's performance, durability, and noise. This study aims to predict the range of dynamic characteristics of the valve train, specifically contact loss and valve velocity.

#### 2.2.1. Contact Loss

Contact loss is a phenomenon that occurs when the valve system, including the tappet and pushrod, deviates from the cam profile designed for the rotational motion of the camshaft at high engine revolutions per minute (RPM). As shown in Figure 2, the green circle indicates the point where contact loss occurs. This phenomenon generates a significant impact force in the valve train components, leading to reduced durability [7].

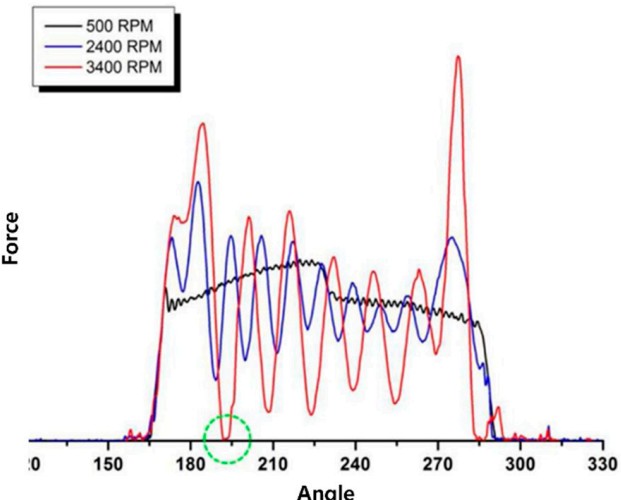

**Figure 2.** Contact loss in the diesel engine valve train.

### 2.2.2. Valve Seating Velocity

Valve seating velocity is defined as the speed at which the valve comes to rest on the valve seat. As depicted in Figure 3, it is necessary to confirm if the valve seating velocity, at the moment the valve lift reaches zero, meets the target speed as indicated by the green circle. When the target speed is exceeded, it adversely affects the valve train noise, the occurrence of valve seat wear, and the durability of the components [7].

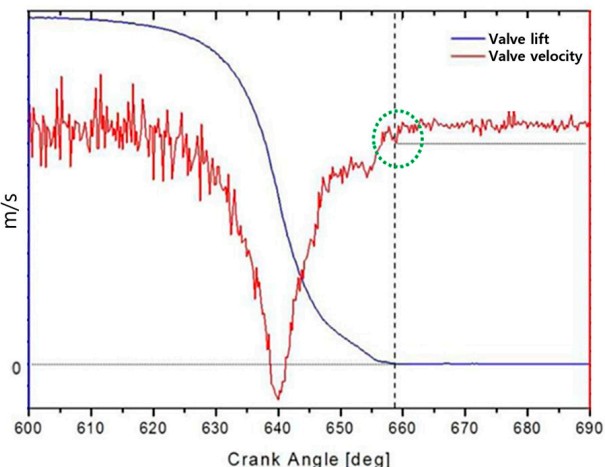

**Figure 3.** Crank angles for valve lift and valve speed.

### 2.3. Related Work

Research on valve trains has been actively conducted in recent years. Hu et al. developed a flexible dynamic model of valve trains by considering the gyroscopic effects and valve gaps, which was achieved through theoretical-experimental analysis of the dynamic properties of valve trains [8]. Similarly, Hu et al. constructed a flexible dynamic model of the valve train system by considering the multi-axis flexibility of the shaft and linkage, and the gyroscopic effect of the camshaft and rotor [9]. While numerical simulation methods are useful for solving the complex nonlinear problems of diesel engines, they have the disadvantage of slowing down the calculation process as the structure of diesel engines becomes more complex. To address this limitation, Zheng et al. explored the application of artificial intelligence algorithms, such as deep neural network-based diesel performance modeling and prediction, combining virtual sample generation (VSG) and a deep neural network (DNN) [10]. However, additional experimental results need to be presented to validate the effectiveness of these approaches. Nonetheless, these studies contribute to an advancement in the understanding of the dynamics of valve trains and can provide insights into the optimization of valve train design and performance.

In recent years, researchers have proposed various innovative approaches for the diagnosis of engine faults. Jiang et al. proposed a method for early warning of abnormal valve clearance in diesel engines using a combination of multi-domain feature extraction and an improved support vector machine (SVM) [11]. Kumar et al. proposed a method for identifying engine faults in two-wheelers using a wavelet synchro-squeezed transform (WSST) and a convolutional neural network (CNN), which outperformed the existing methods [12]. Lastly, Ramteke et al. presented potential fault diagnosis techniques utilizing vibration and acoustic emission analyses, signal processing methods, and artificial neural network models to diagnose scuffing faults on diesel engine components [13].

Recently, many studies have been conducted on engine-related fault and anomaly detection and diagnosis, and Table 1 summarizes the targets and methods of each study. These studies demonstrate the potential of advanced techniques such as deep learning and signal processing for the accurate diagnosis of engine faults.

**Table 1.** The targets and methods of related works.

| Reference | Target | Method |
|---|---|---|
| Hu et al. (2021) [8,9] | Engine valve train | Flexible Dynamic Model |
| Zheng et al. (2021) [10] | Diesel engines simulation | Virtual Sample Generation (VSG) and Deep Neural Network (DNN) |
| Jiang et al. (2019) [11] | Abnormal valve clearance in diesel engines | Support Vector Machine (SVM) with multi-domain feature extraction |
| Kumar et al. (2020) [12] | Engine defects in two-wheelers | Wavelet Synchro-Squeezed Transform (WSST) and Convolution Neural Network (CNN) |
| Ramteke et al. (2022) [13] | Diagnosis and classification of diesel engine components faults | Fast Fourier Transform (FFT), Short-Time Fourier Transform (STFT), and Artificial Neural Network (ANN) |

## 3. Predictive Model

### 3.1. Gradient Boosting Regressor Tree (GBRT)

Boosting is an ensemble machine learning technique that involves combining multiple weak learners to generate a strong learner, as shown in Figure 4. Among the various boosting algorithms, gradient boosting is a highly popular and widely used method that aims to improve the model's prediction accuracy by building on the predictions made by previous models [14].

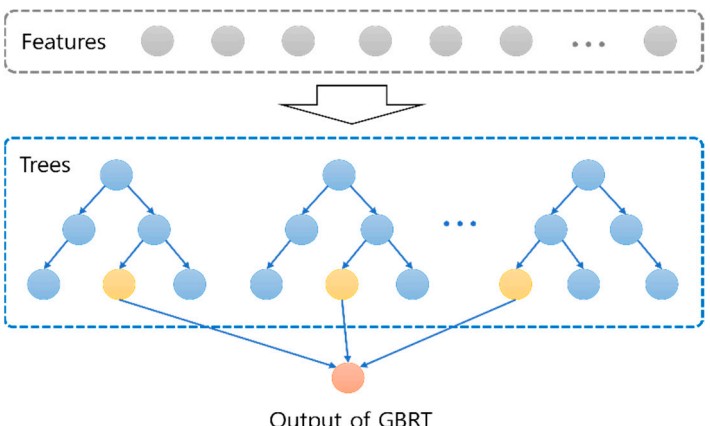

**Figure 4.** The structure of the GBRT model.

The gradient boosting algorithm starts by creating the first model that calculates the average prediction value of the target variables across the entire dataset and computes the residual. This residual is then used to train multiple decision trees that create a stronger model. The iterative process of improving the model continues by obtaining the gradient of the residual and using it to further reduce the residual in the next model.

Gradient boosting has been documented to be highly effective in enhancing the accuracy of machine learning models [15–17]. It can be applied to a broad range of data types and has been extensively used to address regression problems. Therefore, gradient boosting is a robust and powerful ensemble technique that can significantly enhance the prediction accuracy of machine learning models.

### 3.2. Deep Neural Network (DNN)

A deep neural network (DNN) is a type of artificial neural network (ANN) that comprises multiple hidden layers positioned between the input and output layers, as illustrated in Figure 5. The learning process of DNNs involves a repeated error backpropagation procedure that updates weights to minimize the loss function's value via optimization functions such as pure propagation and stochastic gradient descent.

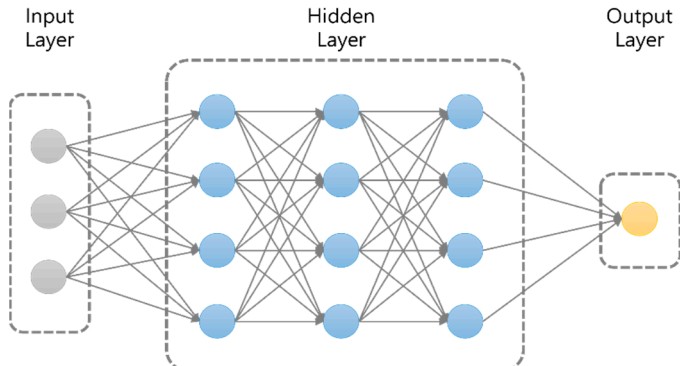

**Figure 5.** The structure of the DNN model.

However, increasing the neural network's depth can lead to the issue of gradient loss, while increasing the number of neurons may result in overfitting. To prevent the loss of inclination, an appropriate weight initialization technique based on the activation function type can be utilized. Overfitting can be prevented by utilizing drop-out and batch normalization techniques. Furthermore, improvements in hardware, such as enhanced graphics processing units (GPUs), have significantly decreased the computation time of complex matrices in deep learning. DNNs that address these limitations can perform complex nonlinear modeling [18]. Therefore, these techniques are highly useful for developing extremely accurate machine learning models that can handle complex, high-dimensional data. In conclusion, DNNs are a powerful tool for addressing complex machine learning problems, and their ability to learn complex non-linear mappings from high-dimensional data makes them very effective in various fields.

### 3.3. One Dimension-Convolution Neural Network (1D-CNN)

Convolutional neural networks (CNNs) are widely used in image and signal processing to simplify model complexity and extract essential features [19,20]. The convolutional kernel, which multiplies the input data with a specific filter in each region and then adds the results, is used to identify distinctive characteristics. As the number of hidden layers increases, the CNN can extract more complex and meaningful features, resulting in improved classification performance.

Although a 2D-CNN kernel is widely used in image classification, the use of a 1D-CNN kernel, as shown in Figure 6, is gaining popularity in other areas such as natural language processing, signal processing, and time series analysis [21,22]. 1D-CNNs can capture important temporal features in data sequences, which makes them well suited for processing time-series data. Additionally, they can be used for feature extraction in natural language processing, where they help to capture local patterns and relations between words.

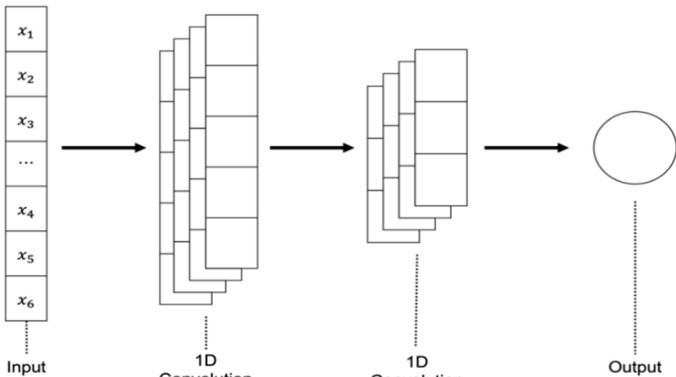

**Figure 6.** The structure of the 1D-CNN model.

### 3.4. Long-Short Term Memory (LSTM)

Long-short term memory (LSTM) is a specialized form of recurrent neural network (RNN) designed to tackle the issue of gradient loss in existing RNNs. The LSTM architecture comprises a hidden layer node, referred to as an LSTM memory cell, which integrates three gates: an input gate, a forget gate, and an output gate, as depicted in Figure 7. The memory cell can erase unnecessary information, update new information, and output pertinent information. The LSTM's strength in retaining long-term memory is crucial in remembering significant features for prolonged periods, which can aid in enhancing performance in time-series-forecasting and language-modeling tasks [23].

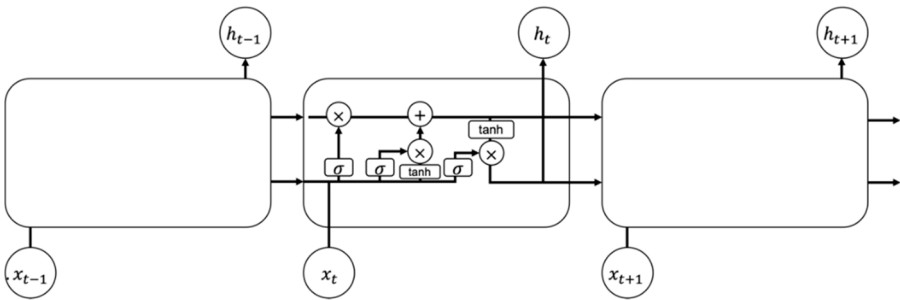

**Figure 7.** The structure of the LSTM model.

## 4. Data Engineering and Model Configuration

### 4.1. Data Collection and Preprocessing

As the use of artificial intelligence (AI) becomes increasingly prevalent, it is important to ensure the reliability and accuracy of AI-based solutions. One critical factor in achieving this goal is the accuracy of the data used to train the AI models. To address this issue, formal techniques can be employed to verify the accuracy and consistency of the gathered and processed data. These techniques offer a systematic approach to detecting and eliminating errors in the data, thereby improving the performance of AI-based solutions. Recent research has demonstrated the effectiveness of such techniques in checking the correctness of AI-based solutions [24,25]. Therefore, incorporating formal techniques into the data-gathering and -processing workflows can enhance the reliability and accuracy of AI-based solutions.

#### 4.1.1. Data Collection

The dataset employed in this research was obtained from AVL's EXCITE Timing Drive Dynamics analysis program [26], which provides dynamic characteristics of valves during the engine development process for three different engine specifications: Base, Marine, and CA. The dataset comprises information on valve dynamic characteristics, as shown in Figure 8. The data are collected during the engine development process, and the program is used to obtain the relevant information.

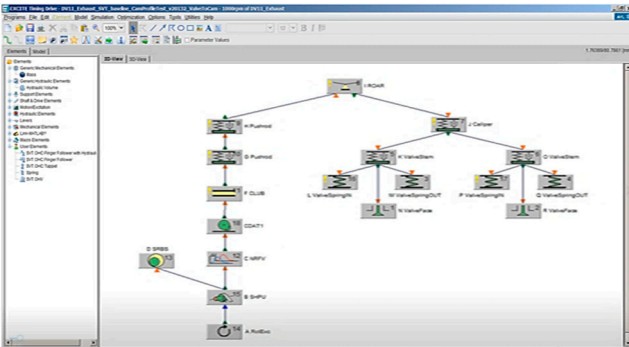

**Figure 8.** Schematic diagram of AVL EXCITE timing drive dynamics.

The dataset utilized in this study is composed of 68,453 rows and 5 columns. A graphical representation of the independent and dependent variables present in the entire dataset is displayed in Figure 9. The dependent variables include time, lift, and CRS angle, while the independent variables comprise valve train force and valve seating velocity.

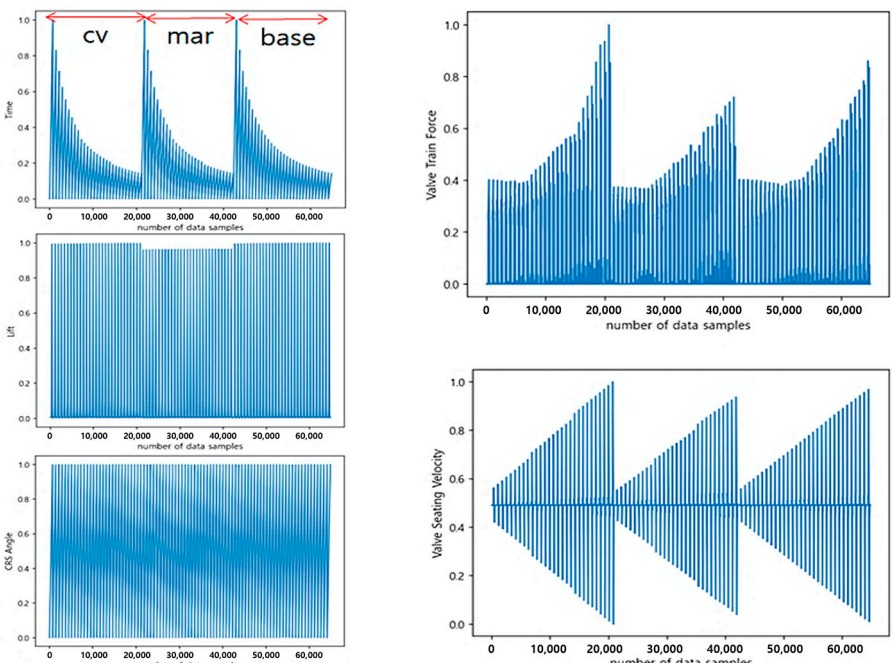

**Figure 9.** The entire dataset in the diesel engine valve train.

### 4.1.2. Data Preprocessing

The data preprocessing process utilized for model learning is presented in Figure 10. Firstly, the valve train dynamic characteristic analysis raw data file for the three CAM profile specifications obtained through the analysis program was imported into Python. A dataset was then generated by adding speed and App (Application) columns to differentiate the engine rotation speed and specifications for each CAM profile. Secondly, to align the lift axes of the three specifications, the maximum value of the lift was examined, and the crank shift (CRS) angle data were processed to match the maximum axes of the lift for each specification. Thirdly, the CRS angle section (row) that is not necessary for learning was removed, and all data were merged. Finally, the Mar and CV specification data were categorized as model learning data, and the Base specification data were categorized as data to be verified in the generated predictive model.

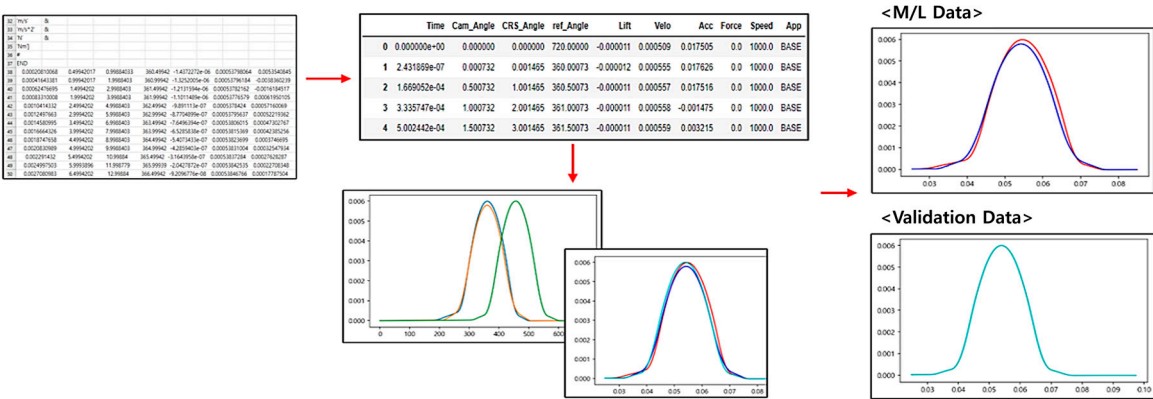

**Figure 10.** The preprocessing process for the diesel engine valve train data.

*4.2. Configuring a Predictive Model*

4.2.1. GBRT Prediction Model

The dataset used for model learning comprises independent variables from the CV and MAR specification datasets, including time, CRS angle, and valve lift, as well as dependent variables such as valve train force and valve seating velocity. The dataset contains a total of 25,073 data points, with a training-to-test ratio of 8:2. The training dataset consisted of 20,059 data points, while the test dataset contained 5014 data points. The optimized parameters, as indicated in Table 2, were set as follows: learning rate = 0.1, max depth = 21, and min sample leaf = 5.

**Table 2.** Hyperparameters used in the GBRT prediction model.

| Parameter | Default | Optimization |
|---|---|---|
| Learning Rate | 0.1 | 0.1 |
| Max Depth | 3 | 21 |
| Min Sample Leaf | 1 | 5 |
| Parameter | Default | Optimization |

4.2.2. Deep Learning-Based Prediction Model

In this study, we employed three different deep learning-based prediction models to analyze the data. The first model we used was a DNN, which served as a basic deep neural network model. Additionally, we utilized an LSTM model, which is specifically designed to overcome the performance degradation that can occur due to the loss of gradient in recurrent neural networks (RNNs). Finally, we employed a 1D-CNN model, which has been shown to perform well on time series-shaped data. The dataset and variables used were the same as those used in the GBRT model. The dataset was normalized using maximum minimum normalization, and the data format was changed from two dimensions to three dimensions to fit the input form of the 1D-CNN and LSTM models. The dimensions were data size, time step, and input dimension, with a time interval of 1 and an input data dimension of 3 (the number of dependent variables). Hyperparameters were selected through a trial-and-error method, as listed in Table 3. The validation split was set to 0.2, and the optimizer was Adam for all models. The number of epochs was 1000 for predicting the valve train force and 30 for predicting the valve seating velocity. The activation function was set as the Tanh function only for the LSTM model. All deep learning-based models had three hidden layers, with 256, 512, and 512 nodes per layer.

**Table 3.** Hyperparameters used in the deep learning-based prediction model.

| Target | Epochs | Validation Split | Activation Function | Optimizer |
|---|---|---|---|---|
| Valve train force | 1000 | 0.2 | ReLU, Tanh(LSTM) | Adam |
| Valve seating velocity | 30 | 0.2 | ReLU, Tanh(LSTM) | Adam |

## 5. Experimental Results

*5.1. Performance Evaluation of Predicted Models*

5.1.1. Evaluation Metrics

This study utilizes valve train system dynamics analysis data obtained during the development of an existing engine. Machine learning techniques, including a gradient boosting regression tree (GBRT), deep neural networks (DNN), and 1D-CNN predict, were employed to construct the predictive model using Python, Scikit-learn and Keras libraries. The predictive performance of the models was evaluated using a decision factor ($R^2$), a mean absolute error (MAE), and a root mean squared error (RMSE), while a separate dataset was used to test and verify the selected model. The test included assessing the performance of each prediction model on a separate dataset and generating graphs to

compare the predicted and actual values of valve train dynamic characteristics such as contact loss and valve seating velocity.

$$R^2 = 1 - \frac{\sum_{i=1}^{n}(X_i - Y_i)^2}{\sum_{i=1}^{n}(X_i - X_{avg})^2} \tag{1}$$

$$MAE = \frac{1}{n}\sum_{i=1}^{n}|X_i - Y_i| \tag{2}$$

$$RMSE = \sqrt{\frac{1}{n}\sum_{i=1}^{n}(X_i - Y_i)^2} \tag{3}$$

5.1.2. GBRT Prediction Model Performance

The performance evaluation of the predictive model is presented in Table 4, where the GBRT model exhibits R-squared values of 0.97 and 0.98, RMSE values of 8.17 and 0.98, and MAE values of 66.84 and 0.011 when predicting the valve train force and valve seating velocity, respectively, based on the test dataset. The prediction results of the GBRT model for the test dataset are visualized in Figures 11 and 12.

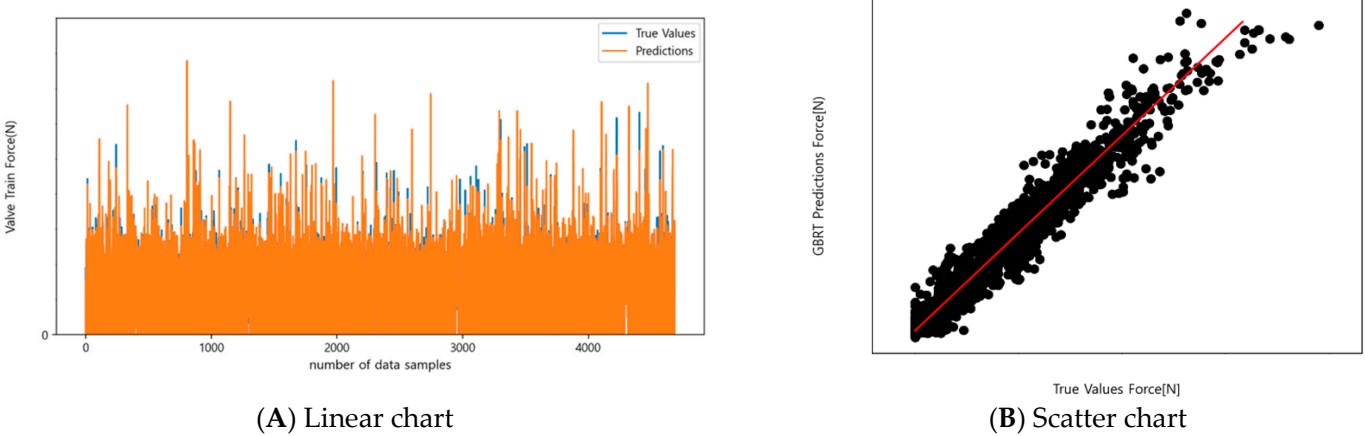

(**A**) Linear chart        (**B**) Scatter chart

**Figure 11.** The predicted values of the valve train force using the GBRT model.

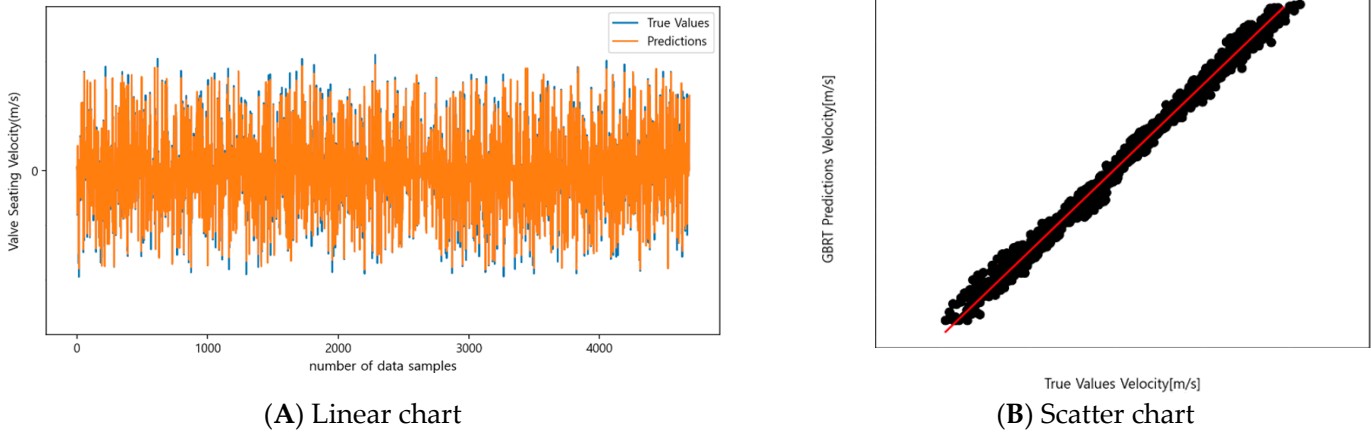

(**A**) Linear chart        (**B**) Scatter chart

**Figure 12.** The predicted values of the valve seating velocity using the GBRT model.

**Table 4.** The performance evaluation results of the GBRT prediction model.

| Metric | $R^2$ | | RMSE | | MAE | |
|---|---|---|---|---|---|---|
| **Target** | **Train** | **Test** | **Train** | **Test** | **Train** | **Test** |
| Valve train force | 0.99 | 0.97 | 67.41 | 120.59 | 37.43 | 66.84 |
| Valve seating velocity | 0.99 | 0.98 | 0.008 | 0.02 | 0.004 | 0.011 |

5.1.3. Performance of Deep Learning-Based Predictive Models

Table 5 displays the evaluation outcomes, indicating that the LSTM model achieved the highest performance for the training dataset, while the 1D-CNN model demonstrated the best overall performance for the test dataset. Figure 13 depicts the prediction results of the valve train force for the test data of the 1D-CNN model. In contrast, in the prediction of valve seating velocity, Table 6 indicates that the 1D-CNN model demonstrated the most favorable outcomes across all performance metrics. Furthermore, Figure 14 showcases the prediction results of the valve seating velocity for the test data of the 1D-CNN model.

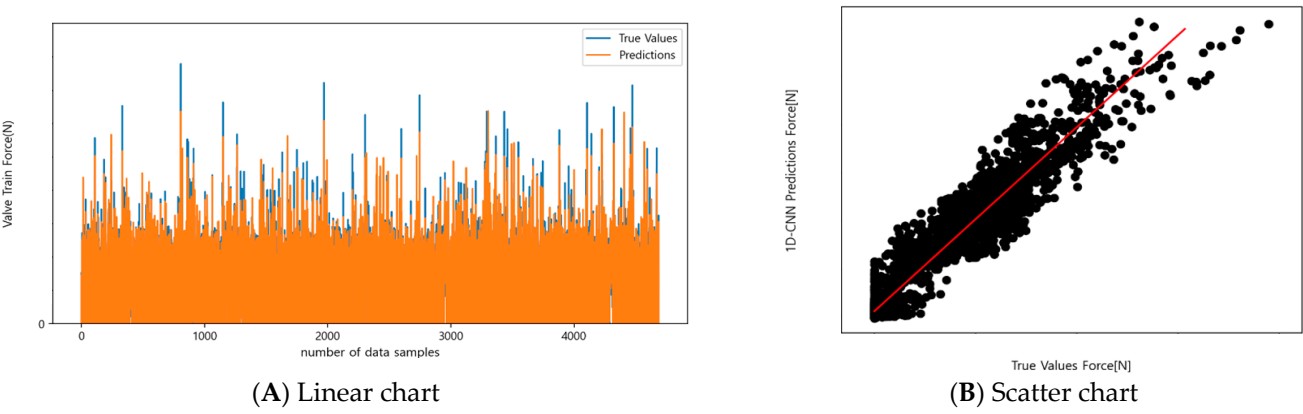

(**A**) Linear chart    (**B**) Scatter chart

**Figure 13.** The predicted values of the valve train force using the 1D-CNN model.

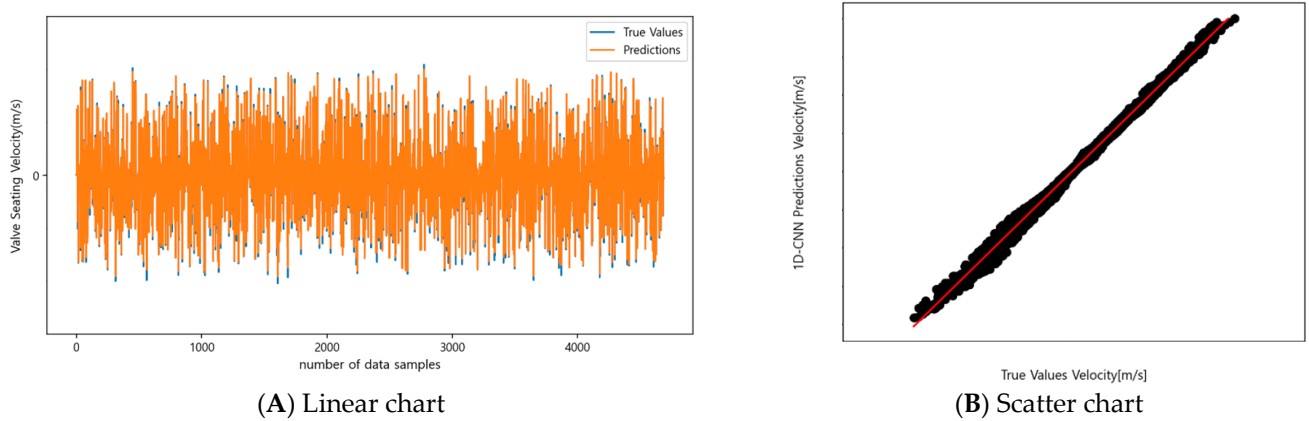

(**A**) Linear chart    (**B**) Scatter chart

**Figure 14.** The predicted values of the valve seating velocity using the 1D-CNN model.

**Table 5.** A performance comparison of prediction models for valve train force prediction.

| Metric | $R^2$ | | RMSE | | MAE | |
|---|---|---|---|---|---|---|
| **Model** | **Train** | **Test** | **Train** | **Test** | **Train** | **Test** |
| DNN | 0.936 | 0.9341 | 174.34 | 176.29 | 108.25 | 108.91 |
| 1D-CNN | 0.9638 | **0.9619** | 131.19 | **135.32** | 73.85 | 77.24 |
| LSTM | **0.9693** | 0.9553 | **120.77** | 146.64 | **56.14** | **72.65** |

**Table 6.** A performance comparison of prediction models for valve seating velocity prediction.

| Metric | $R^2$ | | RMSE | | MAE | |
|---|---|---|---|---|---|---|
| Model | Train | Test | Train | Test | Train | Test |
| DNN | 0.9846 | 0.9839 | 0.0768 | 0.0779 | 0.0043 | 0.0112 |
| **1D-CNN** | **0.9923** | **0.9919** | **0.0530** | **0.0540** | **0.0028** | **0.0029** |
| LSTM | 0.9896 | 0.9893 | 0.0610 | 0.0614 | 0.0037 | 0.0038 |

*5.2. Validate Predictive Models*

5.2.1. Comparison of Model Performance with Base Specification Data

During the construction of the prediction model, a test was conducted on the base specification data, which was excluded from the learning dataset. The actual values and the predicted values were then compared through a performance check and a graph of the prediction model. Among the DNN, 1D-CNN, and LSTM models, the 1D-CNN model with the best predictive performance was selected for comparison with the GBRT model.

Table 7 presents the results of the predictive performance comparison between the GBRT and 1D-CNN models for valve train force. The GBRT model achieved superior performance with an $R^2$ of 0.90, RMSE of 204.19, and MAE of 108.43, while the 1D-CNN model demonstrated an $R^2$ of 0.89, RMSE of 221.44, and MAE of 119.08.

**Table 7.** A comparison of performance for Base specification data of GRBT and 1D-CNN models (valve train force).

| Metric | $R^2$ | RMSE | MAE |
|---|---|---|---|
| Model | Test | Test | Test |
| **GBRT** | **0.90** | **204.19** | **108.43** |
| 1D-CNN | 0.89 | 221.44 | 119.08 |

Furthermore, Table 8 displays the comparison results of the two models for valve seating velocity. In this case, the 1D-CNN model outperformed the GBRT model with an $R^2$ of 0.98, RMSE of 0.07, and MAE of 0.05, whereas the GBRT model exhibited an $R^2$ of 0.97, RMSE of 0.09, and MAE of 0.06.

The graphical representation of the predicted and actual values for the base specification test data of the GBRT model and 1D-CNN model is presented in Figures 15 and 16, respectively.

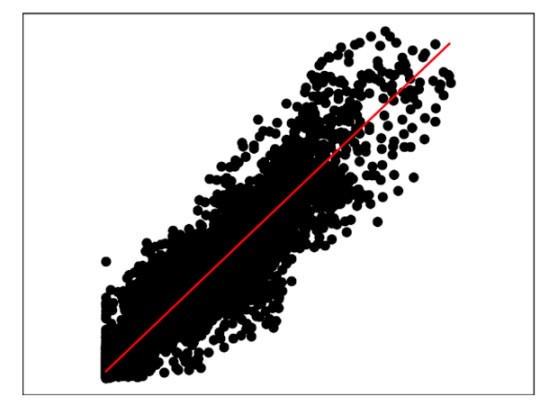

**(A)** The GBRT model

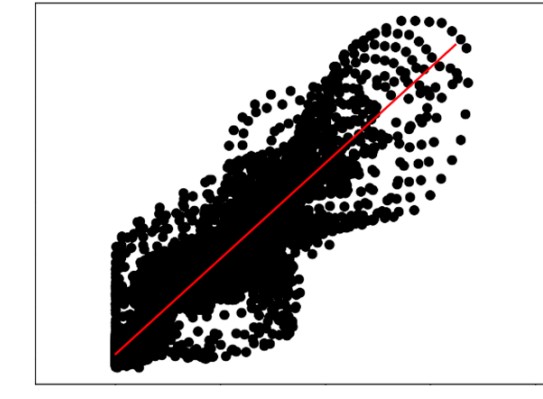

**(B)** The 1D-CNN model

**Figure 15.** Chart comparing the actual and predicted values of the GBRT and 1D-CNN models based on base specification data.

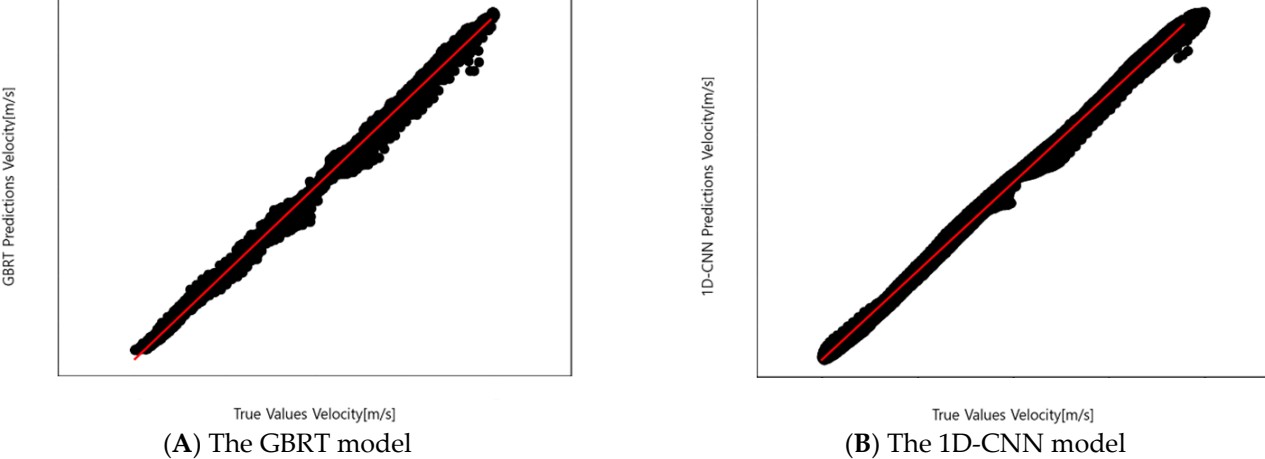

(**A**) The GBRT model  (**B**) The 1D-CNN model

**Figure 16.** Chart comparing the actual and predicted values of the valve train force by diesel engine RPM using the GBRT and 1D convolutional neural network models.

**Table 8.** A comparison of performance for Base specification data of GRBT and 1D-CNN models (valve seating velocity).

| Metric | $R^2$ | RMSE | MAE |
|---|---|---|---|
| Model | Test | Test | Test |
| GBRT | 0.97 | 0.09 | 0.06 |
| **1D-CNN** | **0.98** | **0.07** | **0.05** |

5.2.2. Comparison of Valve Train Force and Valve Seating Velocity Prediction Results

Figure A1 showcases the valve train force based on the crank angle reference engine RPM for both the GBRT and 1D-CNN models, as presented in (A) and (B), respectively. The blue circle indicates the crucial section where the contact loss transpires, which corresponds to the crank angle range of roughly 130 to 180 degrees. Although both models exhibit reasonable accuracy in predicting the valve train force, the 1D-CNN model displays superior consistency with the contact loss interval compared to the GBRT model.

Figure A2 illustrates the valve train force for all engine revolutions as predicted by the GBRT and 1D-CNN models, presented in (A) and (B), respectively. The red circle highlighted in Figure A2 represents the valve train system section where contact loss occurs, resulting in a minimum force of 0. Both predictive models demonstrate similar levels of correspondence with the actual value across the entire range of engine revolutions (1000–3500 RPM).

Figure A3 presents the valve seating velocity as a function of the crank angle reference engine RPM for both the GBRT and 1D-CNN models, represented in (A) and (B), respectively. The blue circle highlights the primary region of interest for the valve seating velocity target speed, which corresponds to the crank angle range of approximately 320 degrees to 350 degrees. Although both predictive models demonstrate good agreement with the actual values, the GBRT model exhibits superior performance for some critical intervals at engine RPM compared to the 1D-CNN model.

Figure A4 displays a comparison graph of the predicted and actual values of the valve seating velocity for the entire engine revolution range (1000–3500 RPM) for both the GBRT and 1D-CNN models, as shown in (A) and (B), respectively. The predictions of both models for the entire engine revolution range are comparable, and fall within the development target value, demonstrating an almost identical level of correspondence with the actual value.

## 6. Discussions and Limitations

The study presented in this paper has several limitations that could be addressed in future research. Firstly, the focus of the study was only on the diesel engine valve train system, and it would be beneficial to expand the research to other engine types and valve train systems to determine if the findings can be generalized to other systems. Another limitation is that the study only considered a limited number of design parameters to predict the dynamic characteristics of the valve train system. Including additional design parameters in the analysis could improve the accuracy of the prediction model.

Additionally, the study only evaluated a few machine learning and deep learning models, and future research could investigate other models or explore ensemble techniques to further improve the predictive performance of the model. Moreover, the study only utilized one type of dynamic analysis data to construct the prediction model. It would be useful to consider different types of data, such as fatigue data or noise data, to further validate the performance and reliability of the model. Furthermore, the study only used a specific dataset, and future research could involve collecting data from more diverse sources to improve the generalization of the prediction model. This could include data from different engine manufacturers, testing conditions, and valve train components.

Therefore, while this study provides a promising foundation for future research into utilizing machine learning and deep learning techniques to predict the dynamic characteristics of valve train systems, more research is needed to address these limitations and validate and improve the predictive performance of the model.

## 7. Conclusions

In this study, the performance of the gradient boosting regressor tree (GBRT) and deep learning models such as the deep neural network (DNN), the one dimension convolutional neural network (1D-CNN), and long short-term memory (LSTM) was evaluated for predicting dynamic characteristics based on diesel engine valve train design parameters. The results showed that the GBRT and deep learning models exhibited good performance in predicting valve train force and valve seating velocity. Both models showed similar results, with the 1D-CNN demonstrating better consistency in predicting contact loss and the GBRT exhibiting better follow-up for valve seating velocity.

However, the deep learning models required more time to learn and most of them had lower predictive performance than the GBRT, even when the number of epochs was set to 1000. Although the predictive performance of the deep learning models could potentially be improved by modifying hyperparameters, it would result in increased computation time and cost, making the GBRT the most suitable model for constructing a prediction model with the current dataset.

In our future works, we aim to expand the dataset by collecting more data with varying specifications, including a wider range of engine rotational speeds, valve spring constants, cam profiles, and valve train component mass and rigidity. By doing so, we can improve the generalization and prediction performance of the prediction model, making it more effective in the real world. Finally, we aim to apply the prediction model to other valve train systems and different types of engines to evaluate its applicability and robustness.

**Author Contributions:** Conceptualization, W.L.; methodology, S.L.; software, T.-Y.J.; validation, S.L; formal analysis, S.L.; investigation, T.-Y.J.; resources, W.L.; data curation, T.-Y.J.; writing—original draft preparation, T.-Y.J.; writing—review and editing, S.L.; visualization, T.-Y.J.; supervision, S.L.; project administration, W.L.; funding acquisition, W.L. All authors have read and agreed to the published version of the manuscript.

**Funding:** This work was supported by Inha University.

**Data Availability Statement:** Not applicable.

**Acknowledgments:** Not applicable.

**Conflicts of Interest:** The authors declare no conflict of interest.

**Abbreviations**

| | |
|---|---|
| AI | Artificial intelligence |
| VSG | Virtual Sample Generation |
| SVM | Support Vector Machine |
| WSST | Wavelet Synchro-Squeezed Transform |
| FFT | Fast Fourier Transform |
| STFT | Short-Time Fourier Transform |
| GBRT | Gradient Boosting Regressor Tree |
| DNN | Deep Neural Network |
| ANN | Artificial Neural Network |
| 1D-CNN | One-Dimensional Convolution Neural Network |
| LSTM | Long Short-Term Memory |
| RNN | Recurrent Neural Network |
| CNN | Convolutional Neural Network |
| GPU | Graphics Processing Unit |
| OHC | Overhead Camshaft |
| OHV | Overhead Valve |
| RPM | Revolutions Per Minute |
| CRS | Crank Shift |
| ReLU | Rectified Linear Unit |
| Tanh | Hyperbolic Tangent |
| MAE | Mean Absolute Error |
| RMSE | Root Mean Squared Error |

**Appendix A**

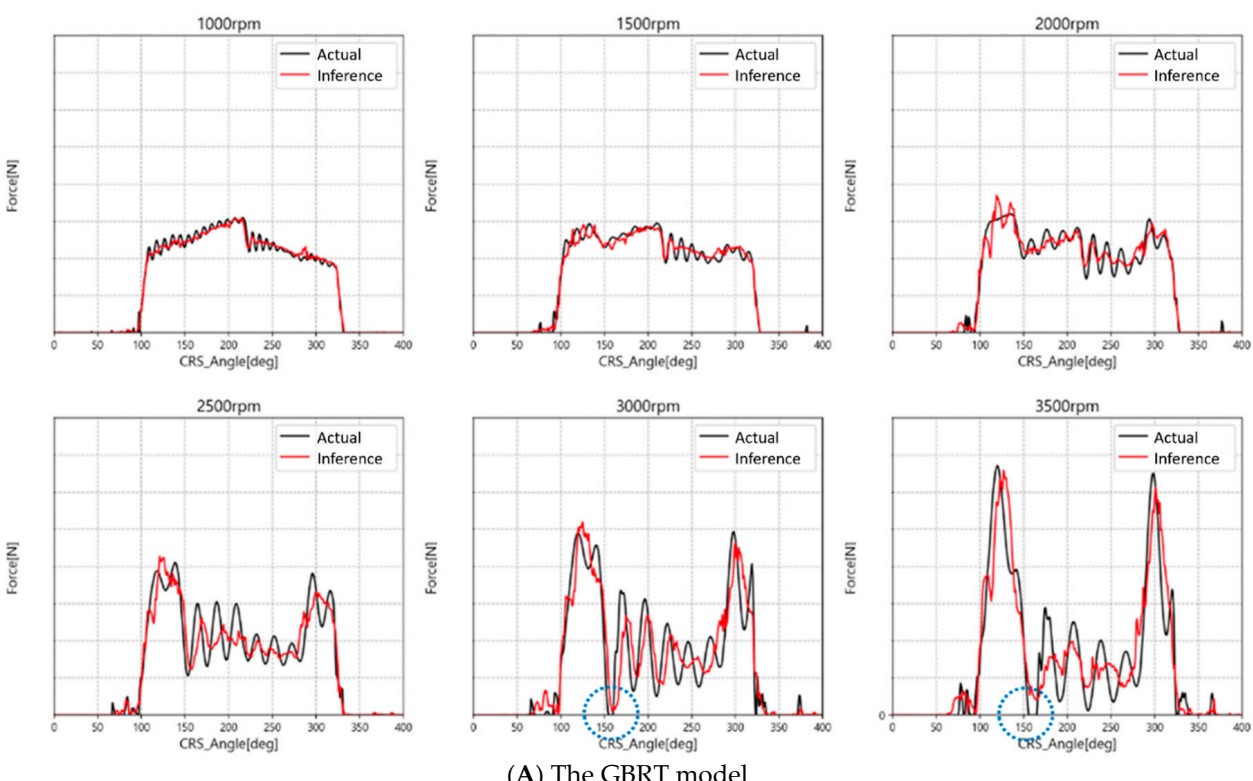

(**A**) The GBRT model

**Figure A1.** *Cont.*

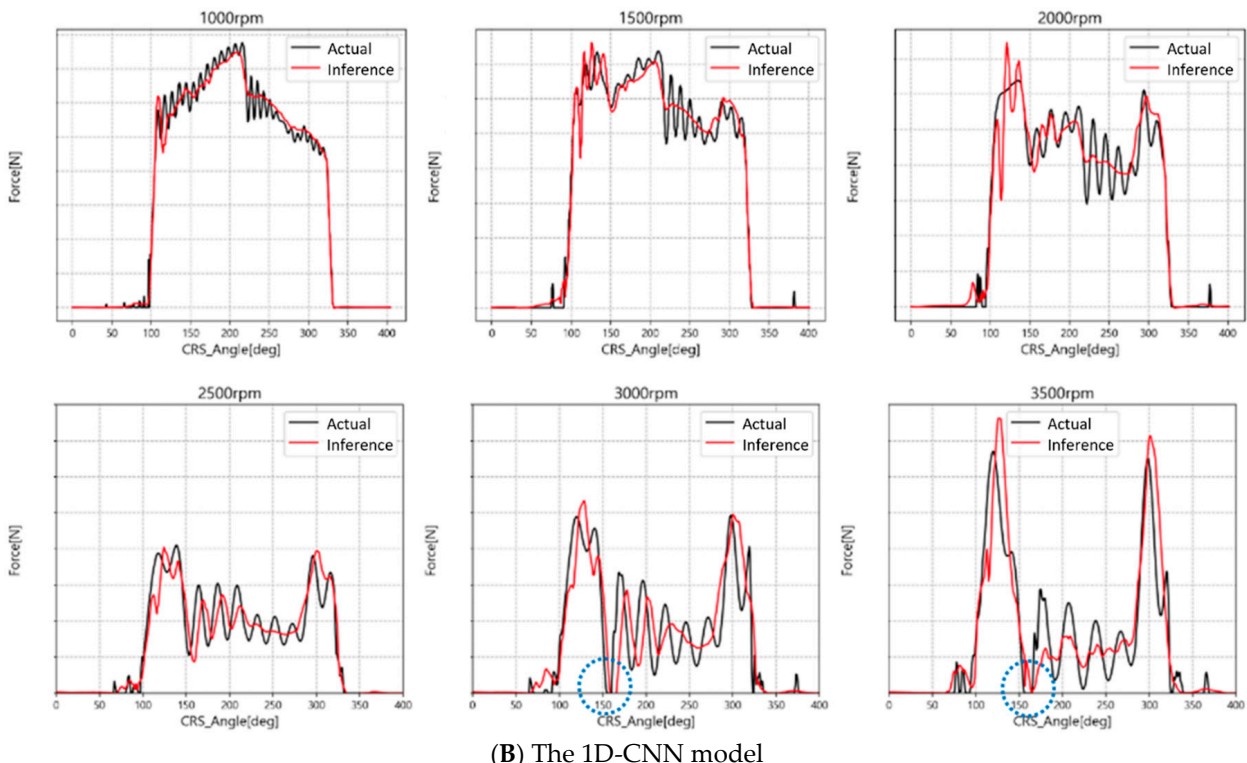

(**B**) The 1D-CNN model

**Figure A1.** Chart comparing the actual and predicted values of the valve train force by crank angle reference RPM using the GBRT and 1D-CNN models.

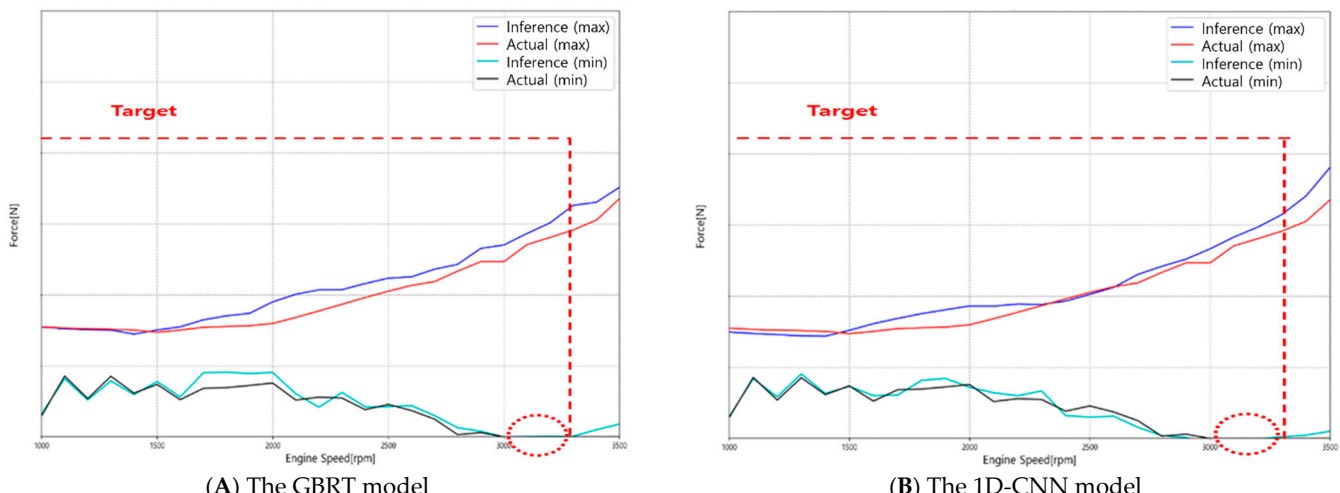

(**A**) The GBRT model                               (**B**) The 1D-CNN model

**Figure A2.** Comparing the predicted values of the valve seating velocity based on the crank angle.

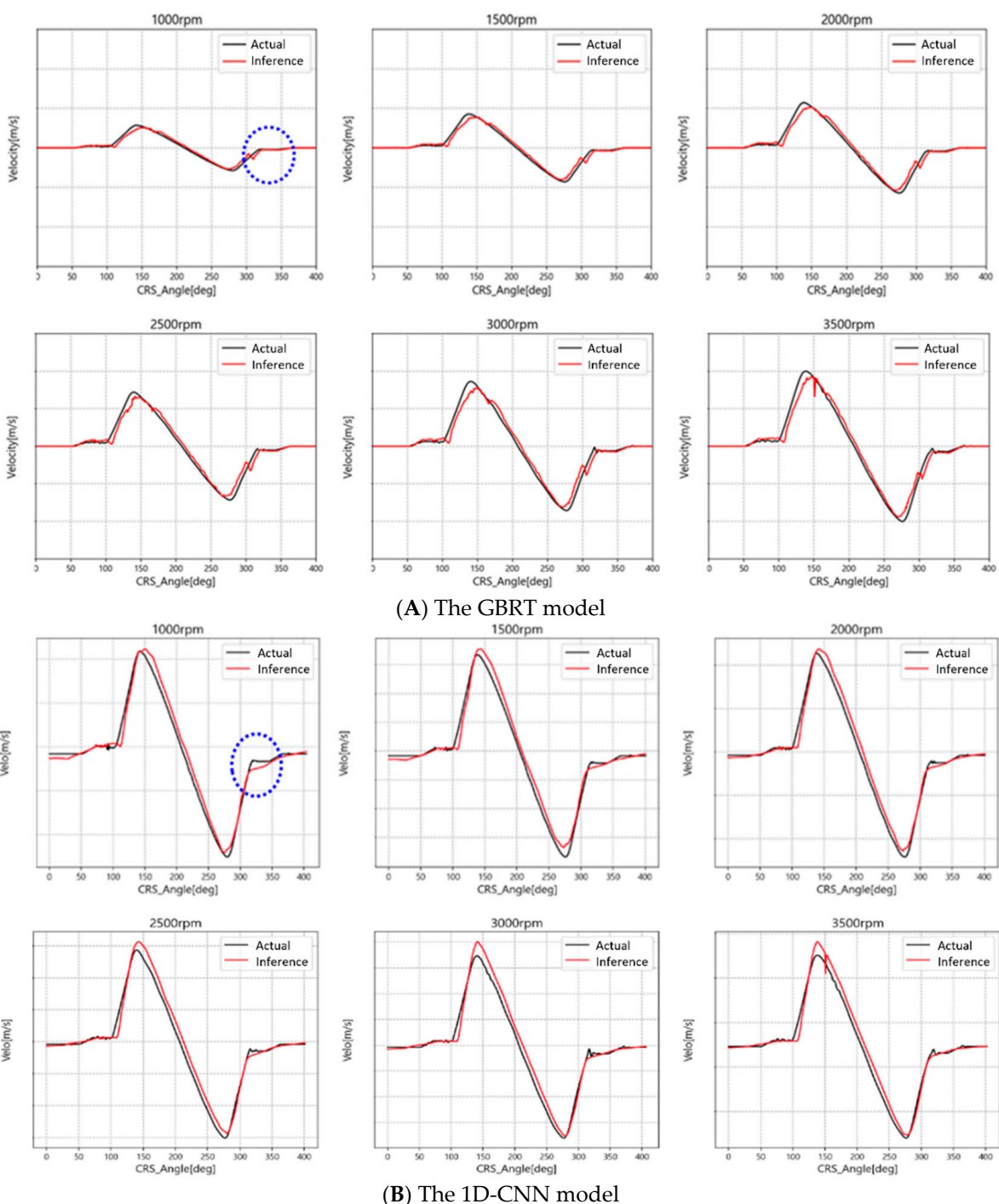

(**A**) The GBRT model

(**B**) The 1D-CNN model

**Figure A3.** Comparing the actual and predicted values of the valve seating velocity by crank angle reference RPM using the GBRT and 1D-CNN models.

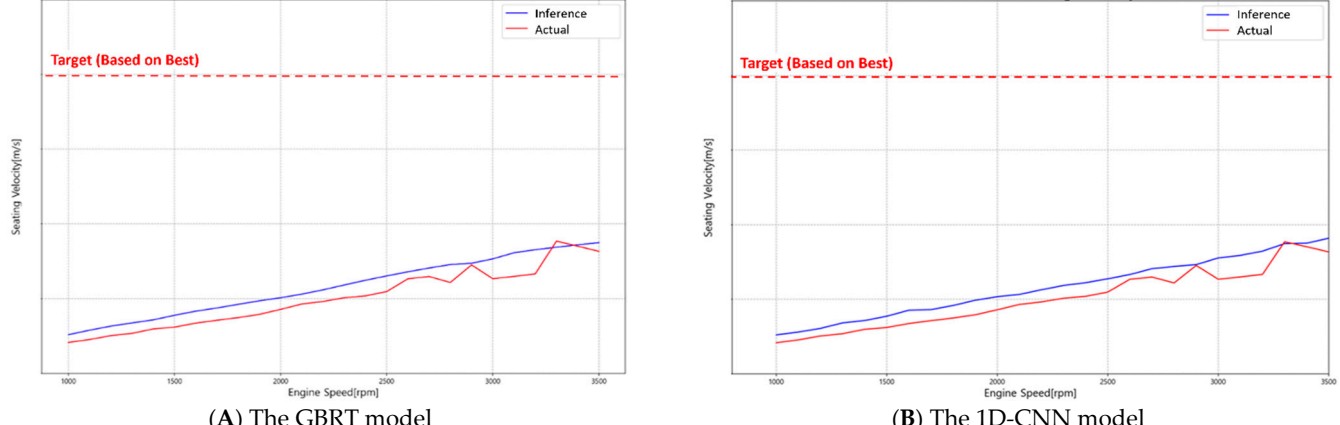

(**A**) The GBRT model  (**B**) The 1D-CNN model

**Figure A4.** Comparing the actual and predicted values of the valve seating velocity throughout engine RPM using the GBRT and 1D-CNN models.

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
