# Peer review of "Dynamic Characteristics Prediction Model for Diesel Engine Valve Train Design Parameters Based on Deep Learning"

_electronics, doi:10.3390/electronics12081806_

Round 1
Reviewer 1 Report
I appreciate your efforts in simplifying a complex work to be digestible for all levels of readers.
Please find my highlighted minor notes on the attached PDF.
Keep up the good work.

Author Response
I appreciate your efforts in simplifying a complex work to be digestible for all levels of readers.
Please find my highlighted minor notes on the attached PDF.
Keep up the good work.
→ Thank you for taking the time to review our paper, and for providing us with valuable feedback. We appreciate your efforts in improving the quality of our work.
We are pleased to inform you that we have carefully considered your comments and have made the necessary revisions to the manuscript. Specifically, we have addressed the typos that you have pointed out and have added axis labels to the figures to enhance the clarity of our presentation. The modified parts have been highlighted in yellow in the revised manuscript for your convenience.
We would like to express our gratitude for your insightful feedback, which has undoubtedly contributed to the improvement of our work.

Reviewer 2 Report
This paper investigated the applications of several machine learning algorithms (GBRT, DNN, 1D-CNN, LSTM) in predicting dynamic characteristics of the design parameters for diesel engine valve train cases, and found that GBRT is the best option considering both predictive performance and development cost. The findings do not provide unique new knowledge as it is well recognized in both academia and industry that tree-based gradient boosting machines (except for the one used in this paper, others examples include XGBOOST, LGBM, etc.) are the top choices for tabular data. However, we should encourage the adoption of ML in the traditional engineering industries, and this paper might be helpful for the audience in the particular industry. Overall, the paper is technically sound and appropriate for publication.
Author Response
This paper investigated the applications of several machine learning algorithms (GBRT, DNN, 1D-CNN, LSTM) in predicting dynamic characteristics of the design parameters for diesel engine valve train cases, and found that GBRT is the best option considering both predictive performance and development cost. The findings do not provide unique new knowledge as it is well recognized in both academia and industry that tree-based gradient boosting machines (except for the one used in this paper, others examples include XGBOOST, LGBM, etc.) are the top choices for tabular data. However, we should encourage the adoption of ML in the traditional engineering industries, and this paper might be helpful for the audience in the particular industry. Overall, the paper is technically sound and appropriate for publication.
→ We appreciate your review of our paper on the applications of machine learning algorithms for predicting dynamic characteristics of design parameters in diesel engine valve train cases. We are pleased to hear that you found our work to be technically sound and appropriate for publication.
We also agree with your observation that the use of tree-based gradient boosting machines for tabular data is a well-established practice in both academia and industry, and that our findings may not provide unique new knowledge in this regard. Nonetheless, we believe that our study can contribute to encouraging the adoption of machine learning techniques in traditional engineering industries, such as the automotive industry.
As you noted, our results indicate that GBRT is the best option in terms of both predictive performance and development cost. This finding can be useful for practitioners in the particular industry, as it provides a practical guideline for selecting the most suitable algorithm for their specific application.
Once again, we appreciate your constructive feedback, and we are glad that our work met your expectations. We hope that our study can serve as a valuable contribution to the field, and we look forward to sharing our findings with a wider audience.

Reviewer 3 Report
This paper presents a comprehensive review on the suitability of machine learning and deep learning approaches in predicting dynamic characteristics of the design parameters with the diesel engine valve train case. Different prediction models are utilized to estimate the force and valve seating velocity values of the valve train system and the GBRT prediction model outperformed the other models. Here are some comments and suggestions for further revision before it can be accepted for publication.
In Section 3.3, it is reported that a 2D-CNN kernel is currently prevalent in image classification, and some examples should be given, such as: such as: doi:10.3390/app12094356; doi:10.1007/s00170-022-10335-8.
The GBRT prediction model has been deemed the most suitable for this application. More in-depth analysis and comparison should be conducted and presented to clarify the advantages of this model.
Overall, this paper is well-written with scientific soundness, and the experimental results are pretty interesting.
Author Response
This paper presents a comprehensive review on the suitability of machine learning and deep learning approaches in predicting dynamic characteristics of the design parameters with the diesel engine valve train case. Different prediction models are utilized to estimate the force and valve seating velocity values of the valve train system and the GBRT prediction model outperformed the other models. Here are some comments and suggestions for further revision before it can be accepted for publication.
→ We appreciate your review of our paper on the suitability of machine learning and deep learning approaches in predicting dynamic characteristics of the design parameters in the diesel engine valve train case. We are glad that you found our work well-written with scientific soundness and interesting experimental results. We also appreciate your valuable comments and suggestions for further improvement.
In Section 3.3, it is reported that a 2D-CNN kernel is currently prevalent in image classification, and some examples should be given, such as: such as: doi:10.3390/app12094356; doi:10.1007/s00170-022-10335-8.
→ Regarding Section 3.3, we agree that providing examples of prevalent 2D-CNN kernels used in image classification would be helpful for readers. We updated the manuscript accordingly and include relevant references, such as doi:10.3390/app12094356 and doi:10.1007/s00170-022-10335-8.
The GBRT prediction model has been deemed the most suitable for this application. More in-depth analysis and comparison should be conducted and presented to clarify the advantages of this model.
→ We also acknowledge your suggestion to conduct a more in-depth analysis and comparison of the GBRT prediction model with other models to clarify its advantages. We incorporated additional analysis and comparison of the models in the revised manuscript to further elucidate the superiority of the GBRT model. In Section 5.2 of the paper, a comparative analysis of two machine learning methods, namely Gradient Boosting Regression Trees (GBRT) and one-dimensional Convolutional Neural Networks (1D-CNN), was performed. The results of the analysis are presented in Tables 6-7 and Figures 15-20. Based on these findings, the advantages of each method were assessed and discussed. These results were used to provide insight into the relative strengths and weaknesses of each method, which can inform future research and development efforts in this area.
Overall, this paper is well-written with scientific soundness, and the experimental results are pretty interesting.
→ Once again, we appreciate your valuable feedback, and we are committed to improving the quality of our work. We hope that the revised manuscript will be deemed suitable for publication, and we look forward to hearing from you soon.

Reviewer 4 Report
Summary/Contribution: This research provides a comprehensive analysis of the feasibility of machine learning and deep learning algorithms in predicting dynamic design parameters using the diesel engine valve train case. Comments/Suggestions: 1. A brief introduction to the topic under consideration and its significance could be added to the abstract. 2. Abstract: "1D-CNN, and LSTM" ===> Please explain the meaning of these abbreviations as you did for other abbreviations. 3. Please insert some numerical experimental results in the abstract. 4. The authors may add a table at the end of the paper to give the meaning of all used abbreviations. 5. Add a list of short sentences in the introduction that summarize the contributions of the authors in this work. 6. Please write a short paragraph at the end of the introduction which describes the structure of the paper. 7. Section 2.3 (Related Works) needs to be summarized in tabular form in order to emphasize the originality of the proposed work. 8. Figure 4 is too simplistic and needs to be enriched with more details. 9. Figure 8 is of low quality and needs improvement. 10. The paper contains many figures. Some of them may be moved to the appendix. 11. The authors are asked to add a short paragraph about how formal techniques can be used to check the correctness of AI-based solutions, especially when it comes to gathering and processing data. 12. For this purpose, the following references may be included: a. https://ieeexplore.ieee.org/document/9842406 b. https://dl.acm.org/doi/abs/10.1145/3503914 13. Line 220: "The deep learning-based prediction models employed in this study are DNN" ====> The authors need to argue more about this choice. 14. The authors need to identify the limitations of their work and propose more future work directions.
Author Response
A brief introduction to the topic under consideration and its significance could be added to the abstract.
→ Thank you for providing feedback on our paper's abstract. We appreciate the opportunity to improve our work and have made a number of changes to the abstract to provide readers with a clearer understanding of our research. We hope that these changes will address any concerns and help to strengthen the overall impact of our study.
This paper presents a comprehensive study on the utilization of machine learning and deep learning techniques to predict the dynamic characteristics of design parameters, exemplified by a diesel engine valve train. The research aims to address the challenging and time-consuming analysis required to optimize the performance and durability of valve train components, which is influenced by numerous factors.
Abstract: "1D-CNN, and LSTM" ===> Please explain the meaning of these abbreviations as you did for other abbreviations.
→ As per your feedback, we have added explanations of the abbreviations 1D-CNN and LSTM used in the abstract.
To this end, dynamic analyses data have been collected for diesel engine specifications and used to construct a regression prediction model using Gradient Boosting Regressor Tree (GBRT), Deep Neural Network (DNN), One-Dimensional Convolution Neural Network (1D-CNN), and Long Short-Term Memory (LSTM).
Please insert some numerical experimental results in the abstract.
→ We have incorporated quantitative experimental findings into the abstract section of our paper.
The prediction model was utilized to estimate the force and valve seating velocity values of the valve train system. The dynamic characteristics of the case have been evaluated by comparing the actual and predicted values. The results showed that the GBRT model had an R2 value of 0.90 for Valve Train Force and 0.97 for Valve Seating Velocity, while the 1D-CNN model had an R2 value of 0.89 for Valve Train Force and 0.98 for Valve Seating Velocity. The results of this study have important implications for advancing the design and development of efficient and reliable diesel engines.
The authors may add a table at the end of the paper to give the meaning of all used abbreviations.
→ We would like to express our sincere gratitude for taking the time to review our paper. We have carefully considered your feedback and implemented your suggestions. In particular, we have taken your advice to explain the meaning of all the acronyms used at the end of the paper.
To address this concern, we have included a section titled "List of Acronyms" at the end of our paper. This section provides a comprehensive list of all the acronyms used throughout the manuscript, along with their respective meanings. We believe that this addition will help readers who may not be familiar with the terminologies used in the paper, to understand the content more easily.
Abbreviations
AI Artificial intelligence
VSG Virtual Sample Generation
SVM Support Vector Machine
WSST Wavelet Synchro-Squeezed Transform
FFT Fast Fourier Transform
STFT Short-Time Fourier Transform
GBRT Gradient Boosting Regressor Tree
DNN Deep Neural Network
ANN Artificial Neural Network
1D-CNN One-Dimensional Convolution Neural Network
LSTM Long Short-Term Memory
RNN Recurrent Neural Network
CNN Convolutional Neural Network
GPU Graphics Processing Unit
OHC Overhead Camshaft
OHV Overhead Valve
RPM Revolutions Per Minute
CRS Crank Shift
ReLU Rectified Linear Unit
Tanh Hyperbolic Tangent
MAE Mean Absolute Error
RMSE Root Mean Squared Error
Add a list of short sentences in the introduction that summarize the contributions of the authors in this work.
→ We greatly appreciate your feedback on our paper and have carefully considered your suggestions. In response, we have added descriptions of the paper's main contributions to ensure that our readers have a clear understanding of the novel aspects of our work.
The main contributions of this paper are as follows:
- We compared various machine learning and deep learning models such as GBRT, DNN, 1D-CNN, and LSTM to predict the dynamic characteristics of diesel engine valve train design parameters.
- We present an account of the data preprocessing process, which involved im-porting raw data, creating a dataset with speed and application columns, aligning lift axes, removing unnecessary data, and classifying specifications for model training and validation.
- We demonstrate the validity and effectiveness of this study by performing a detailed comparative analysis of models predicting valve train force and valve seating velocity over a range of crankshaft angles and engine speeds.
Please write a short paragraph at the end of the introduction which describes the structure of the paper.
→ Thank you for providing your feedback on the structure of our paper. In response to your comments, we have included an explanation of the paper's organization to provide greater clarity to our readers. The revised explanation of the paper's structure can be found in the introduction. We hope that this addition will address your concerns and improve the overall readability and comprehension of the paper.
The structure of this paper is outlined as follows. In Section 2, the background and related work is reviewed. Section 3 describes the prediction model in detail. Section 4 de-tails the data engineering and model construction, while Section 5 presents the experimental analysis results. Section 6 describes the discussion and limitations of this paper, Finally, Section 7 concludes the paper.
Section 2.3 (Related Works) needs to be summarized in tabular form in order to emphasize the originality of the proposed work.
→ We would like to express our appreciation for your valuable feedback on our paper. We have taken your suggestion and included a table summarizing the relevant studies, highlighting the originality and importance of our work. The table provides a brief overview of the studies conducted by various researchers in the field of engine fault diagnosis.
Table 1. The targets and methods of related works.
|
Reference |
Target |
Method |
|
Hu et al. (2021) [8-9] |
Engine valve train |
Flexible Dynamic Model |
|
Zheng et al. (2021) [10] |
Diesel engines simulation |
Virtual Sample Generation (VSG) and Deep Neural Network (DNN) |
|
Jiang et al. (2019) [11] |
Abnormal valve clearance in diesel engines |
Support Vector Machine (SVM) with multi-domain feature extraction |
|
Kumar et al. (2020) [12] |
Engine defects in two-wheelers |
Wavelet Synchro-Squeezed Transform (WSST) and Convolution Neural Network (CNN) |
|
Ramteke et al. (2022) [13] |
Diagnosis and classification of diesel engine components faults |
Fast Fourier Transform (FFT), Short-Time Fourier Transform (STFT), and Artificial Neural Network (ANN) |
Figure 4 is too simplistic and needs to be enriched with more details.
→ Thank you for your comment on our paper. We redrew Figure 4 and put it on the paper.
Figure 8 is of low quality and needs improvement.
→ Thank you for your valuable feedback on our paper. We have made revisions to the figure based on your comments.
The paper contains many figures. Some of them may be moved to the appendix.
→ We would like to express our gratitude for your constructive feedback on our paper. We have implemented your suggestions and added Appendix A and moved some of the figures around accordingly. We believe that the addition of these appendices will provide readers with a more detailed understanding of our experimental setup and data analysis methods, which will enhance the clarity and impact of our work. Furthermore, we have moved some of the figures around to better align with the flow of the manuscript. This change will ensure that readers can better understand the results presented in the manuscript.
The authors are asked to add a short paragraph about how formal techniques can be used to check the correctness of AI-based solutions, especially when it comes to gathering and processing data.
→ We greatly appreciate your feedback on our paper and have carefully considered your suggestions. We agree that the accuracy of the data used to train the AI models is crucial for achieving reliable and accurate solutions. Therefore, we have added a short paragraph discussing how formal techniques can be used to verify the correctness of AI-based solutions, especially in data gathering and processing. Formal techniques provide a systematic approach to detect and eliminate errors in the data, which can significantly improve the performance of AI-based solutions.
As the use of Artificial Intelligence (AI) becomes increasingly prevalent, it is important to ensure the reliability and accuracy of the AI-based solutions. One critical factor in achieving this goal is the accuracy of the data used to train the AI models. To address this issue, formal techniques can be employed to verify the accuracy and consistency of the gathered and processed data. These techniques offer a systematic approach to detecting and eliminating errors in the data, thereby improving the performance of AI-based solutions. Recent research has demonstrated the effectiveness of such techniques in checking the correctness of AI-based solutions [24, 25]. Therefore, incorporating formal techniques into the data gathering and processing workflows can enhance the reliability and accuracy of the AI-based solutions.
For this purpose, the following references may be included: a. https://ieeexplore.ieee.org/document/9842406 b. https://dl.acm.org/doi/abs/10.1145/3503914
→ Thank you for your feedback. We have updated our paper as per your suggestion by including the references you pointed out and updating the text accordingly. We believe that the inclusion of these references will strengthen our paper and provide a more comprehensive discussion of the topic. We appreciate your valuable feedback and input and are grateful for the opportunity to improve our work.
Line 220: "The deep learning-based prediction models employed in this study are DNN" ====> The authors need to argue more about this choice.
→ Thank you for your valuable feedback on our paper. We appreciate your comment regarding the deep learning-based prediction models used in this study and would like to provide further justification for our choice of models.
We selected the DNN model as a basic prediction model due to its versatility in handling various types of data and its ability to effectively process large amounts of information. Additionally, the LSTM model was employed to overcome the issue of performance degradation that can occur due to the loss of gradient in recurrent neural networks (RNNs). This model is specifically designed to handle sequential data and has been shown to be effective in various applications. Finally, we utilized the 1D-CNN model, which has been demonstrated to perform well on time series-shaped data.
By using these three different models, we were able to compare their performances and identify the most effective approach for our specific data set. We believe that the use of multiple models in our study provides a comprehensive analysis and enhances the validity of our findings.
Here's what we added to the paper based on your feedback.
In this study, we employed three different deep learning-based prediction models to analyze the data. The first model we used was a DNN, which served as a basic deep neural network model. Additionally, we utilized an LSTM model, which is specifically designed to overcome the performance degradation that can occur due to the loss of gradient in recurrent neural networks (RNNs). Finally, we employed a 1D-CNN model, which has been shown to perform well on time series-shaped data.
The authors need to identify the limitations of their work and propose more future work directions.
→ Thank you for taking the time to review our paper. We appreciate your thoughtful comments and suggestions, which have helped us improve the quality of our research.
We agree with your assessment regarding the need for additional discussion on the limitations of our study, and we have addressed this in a new section.
- Discussions and Limitations
The study presented in this paper has several limitations that could be addressed in future research. Firstly, the focus of the study was only on the diesel engine valve train system, and it would be beneficial to expand the research to other engine types and valve train systems to determine if the findings can be generalized to other systems. Another limitation is that the study only considered a limited number of design parameters to predict the dynamic characteristics of the valve train system. Including additional design parameters in the analysis could improve the accuracy of the prediction model.
Additionally, the study only evaluated a few machine learning and deep learning models, and future research could investigate other models or explore ensemble techniques to further improve the predictive performance of the model. Moreover, the study only utilized one type of dynamic analysis data to construct the prediction model. It would be useful to consider different types of data, such as fatigue data or noise data, to further validate the performance and reliability of the model. Furthermore, the study only used a specific dataset, and future research could involve collecting data from more di-verse sources to improve the generalization of the prediction model. This could include data from different engine manufacturers, testing conditions, and valve train components.
Therefore, while this study provides a promising foundation for future research on utilizing machine learning and deep learning techniques to predict the dynamic characteristics of valve train systems, more research is needed to address these limitations and validate and improve the predictive performance of the model.
We also agree with your suggestions for future research, and we have expanded on this in our Conclusion section.
In our future works, we aim to expand the dataset by collecting more data with varying specifications, including a wider range of engine rotational speeds, valve spring constants, cam profiles, and valve train component mass and rigidity. By doing so, we can improve the generalization and prediction performance of the prediction model, making it more effective in the real world. Finally, we aim to apply the prediction model to other valve train systems and different types of engines to evaluate its applicability and robustness.

Round 2
Reviewer 4 Report
The authors considered my comments and suggestions. Good luck.